# Ultrasound-guided puncture reduces bleeding-associated complications, regardless of calcified plaque, after endovascular treatment of femoropopliteal lesions, especially using the antegrade procedure: A single-center study

**Kentaro Fukuda**[1], **Shinya Okazaki**[2]*, **Masayuki Shiozaki**[1], **Iwao Okai**[2], **Akihisa Nishino**[3], **Hiroshi Tamura**[1], **Kenji Inoue**[1], **Masataka Sumiyoshi**[1], **Hiroyuki Daida**[2], **Tohru Minamino**[2]

**1** Department of Cardiology, Juntendo University Nerima Hospital, Nerimaku, Tokyo, Japan, **2** Department of Cardiovascular Biology and Medicine, Juntendo University Graduate School of Medicine, Tokyo, Japan, **3** Department of Cardiology, Nishino Naika Clinic, Yamanashi, Japan

* shinya@juntendo.ac.jp

**Data Availability Statement:** All relevant data are within the manuscript.

## Abstract

### Background

A common complication of endovascular treatment for femoropopliteal lesions is bleeding at the vascular access site. Although risk factors of bleeding-associated complications at the approach site have been reported, the results have been inconclusive. Hence, this study aimed to assess the predictors of bleeding-associated complications at the approach site in patients undergoing endovascular treatment for femoropopliteal lesions.

### Methods

This retrospective, single-center, observational study included consecutive patients who underwent endovascular treatment (n = 366, 75% male, 72.4±9.9 year) for peripheral arterial disease with claudication and critical limb ischemia in our hospital from January 2010 to December 2017. We divided the patients into bleeding and non-bleeding groups, depending on whether bleeding-associated complications occurred at the approach site. Bleeding-associated complications were defined according to the Bleeding Academic Research Consortium criteria types 2, 3, and 5.

### Results

Altogether, 366 endovascular treatment procedures and 404 arterial accesses were performed for femoropopliteal lesions in 335 peripheral arterial disease patients with claudication and 69 critical limb ischemia patients. We recorded 35 postprocedural bleeding-associated complications at the approach site (9%), all of which were hematomas. The predictors of increased bleeding-associated complications were age ≥ 80 years (bleeding vs.

**Funding:** The author(s) received no specific funding for this work.

**Competing interests:** The authors have declared that no competing interests exist.

non-bleeding group, 43% vs. 25%, p<0.05) and antegrade cannulation of the common femoral artery (48% vs. 69%, p<0.05). Ultrasound-guided puncture reduced bleeding-associated complications (odds ratio, 0.28; 95% confidence interval, 0.004–0.21; p<0.05). In contrast, there was no significant difference in puncture site calcification between the groups (bleeding vs. non-bleeding groups, 29% vs. 21%, p = 0.29).

## Conclusion

Ultrasound-guided puncture is associated with a decrease in bleeding-associated complications at the approach site, regardless of the presence of calcified plaque. It is particularly effective and should be more actively used in patients aged ≥80 years and for antegrade cannulation of the common femoral artery.

## Introduction

Endovascular treatment (EVT) for femoropopliteal (FP) lesions is currently one of the most rapidly expanding fields of medicine. According to the treatment recommendations for FP lesions in 2010 upon this study's initiation, EVT was only recommended for single stenosis of <3 cm in patients with peripheral arterial disease (PAD) with claudication, and surgical intervention was the first choice for patients with other lesions and critical limb ischemia (CLI) [1]. In the past decade, new EVT-related devices and technologies have been introduced sequentially, and improvements in treatment results have been reported [2–7]. Regarding treatment recommendations for FP lesions in 2017 upon this study's completion, EVT was the first choice for stenoses and occlusions of <25 cm in patients with claudication and CLI [8]. With this rapid development of EVT, treating more severe and advanced lesions became possible. In the future, EVT is expected to be developed further and treat more complicated cases than was previously possible. Conversely, complications have become more problematic than previously reported because high-risk patients are currently being targeted, and the reduction in complications is important.

One of the most common complications of EVT is bleeding complications at the vascular access site, with an incidence of 1–10% [9–11]. Identified risk factors include increasing age, the female sex, anticoagulant use, puncture without ultrasonographic guidance, a large sheath size, and manual compression [12–18]. Although various studies have reported on the risk factors, the results are inconclusive. Additionally, in previous studies showing that ultrasound-guided puncture is useful, many of the cases are of retrograde approaches. In modern EVT, the antegrade approach is often required [19], and the purpose was to retroactively confirm whether ultrasound guidance is effective in the recent EVT situation. Hence, this study aimed to examine the risk factors for bleeding complications in EVT of FP lesions using real clinical data from our institution.

## Materials and methods

### Study design and patients

This was a retrospective, single-center study. Altogether, 366 consecutive patients who underwent EVT for FP lesions at our institution from January 2010 to December 2017 were enrolled in this study. We divided the patients into the bleeding group (BG) for those with bleeding-associated complications and the non-bleeding group (NBG) for those without bleeding-

associated complications, and examined the risk factors related to the complications. The study was conducted in accordance with the Helsinki Declaration of 1971, as revised in 1982. Written informed consent was obtained from all patients.

### Inclusion and exclusion criteria

We included all patients who underwent EVT for PAD with claudication and CLI. Patients who underwent EVT performed through antegrade or retrograde common femoral artery (CFA), and popliteal artery or distal below-the-knee artery accesses were included, whereas those who underwent EVT using other accesses, such as the radial or brachial artery, were excluded.

### Endpoints

We reviewed patient demographics, medical comorbidities, and operative details. The main outcome measure of bleeding-associated complications was defined as bleeding at the puncture site that required medical intervention (requiring compression, an extension of the bed rest period, transfusion, and operation for hemostasis), classified as Bleeding Academic Research Consortium criteria types 2, 3, and 5 [20].

### Study participants and clinical assessment

PAD with claudication was diagnosed based on clinical symptoms of intermittent claudication and angiographic findings, which showed >75% stenosis and/or a translational pressure gradient (>20 mmHg) and was classified according to the Trans-Atlantic Inter-Society Consensus (TASC). Meanwhile, CLI was diagnosed based on clinical symptoms of ischemic pain at rest or the presence of tissue loss, such as in PAD-related non-healing ulcers or gangrene. Calcification at the puncture site was defined as a clear confirmation of calcification on a fluoroscopic image.

### Endovascular intervention

Arterial puncture procedures were performed only by certified doctors. The needle aimed at the distal third of the femoral head, and only the anterior wall of the artery was punctured. Patients with normal renal function underwent angiography with iopamidol-370. Those with creatinine level >1.5 mg/dL underwent angiography with carbon dioxide. Heparin (5000 IU) was used in all patients to induce systemic anticoagulation. When the wire passed the culprit lesion, pre-dilation with either a plain balloon or scoring balloon angioplasty was performed. The balloon size was determined by the operator and was based on the angiographic appearance of the vessel. FP lesions were first treated with balloon angioplasty at a nominal pressure for 5 min, and a nitinol stent was introduced when major dissection with contrast delay occurred. In FP lesions, EVT success was defined as residual lesions <50%, without the occurrence of flow-limiting dissection. Protamine was used unless there was a known contraindication such as an allergy. If the puncture position did not involve the bifurcation, hemostasis was attempted using vascular closure devices. In either case, manual compression was performed for 15 min or more by a clinical fellow until hemostasis was achieved, and finally, mechanical compression was added using a tourniquet band. The patient was rested on a bed for 6 h or more after the operation, and walking was permitted after examination of the puncture site the next day.

## Statistical analysis

Continuous variables are presented as mean±SD or median (interquartile range); categorical variables are presented as numbers and percentages. P-values <0.05 are considered statistically significant. Categorical variables are described as numbers and proportions and were analyzed using the chi-square test, whereas continuous variables were analyzed using the Student's *t*-test. Multivariate analysis was performed using variables that were reportedly associated with access site complications. A stepwise multiple regression analysis was performed to determine the variables. All statistical analyses were performed using SPSS version 16.0 (SPSS Inc., Chicago, IL, USA) and JMP version 9.0.0. (SAS Institute Inc., Cary, NC, USA).

## Results

Altogether, 374 EVT procedures were performed from January 2010 to December 2017. The patients' ages ranged from 43 to 95 years (mean: 72.4±9.9 years). Among them, 303 were men (75%), 108 were ≥80 years (27%), 335 had PAD with claudication (83%), 69 had CLI (17%), 355 had hypertension (88%), 308 had dyslipidemia (76%), 250 had diabetes mellitus (62%), and 114 had undergone hemodialysis (28%) (Table 1).

**Table 1. Patients' characteristics.**

| Variables | Total | Bleeding group | Non-bleeding group | p-value |
|---|---|---|---|---|
| | n = 404 | n = 35 | n = 369 | |
| Male | 303 (75) | 26 (74) | 277 (75) | 0.92 |
| Age, years | 72.4±9.9 | 74.7±9.0 | 72.2±9.9 | |
| ≥80 years | 108 (27) | 15 (43) | 93 (25) | <0.05 |
| Diagnosis | | | | 0.35 |
| PAD | 335 (83) | 31 (89) | 304 (82) | |
| CLI | 69 (17) | 4 (11) | 65 (18) | |
| Risk factor | | | | |
| Hypertension | 355 (88) | 28 (80) | 327 (89) | 0.14 |
| Dyslipidemia | 308 (76) | 28 (80) | 280 (76) | 0.58 |
| Diabetes mellitus | 250 (62) | 18 (51) | 232 (63) | 0.18 |
| Hemodialysis | 114 (28) | 5 (14) | 109 (30) | 0.06 |
| Medication | | | | |
| SAPT | 16 (4) | 3 (9) | 13 (4) | 0.14 |
| DAPT | 335 (83) | 27 (77) | 308 (83) | 0.34 |
| SAPT+OAC | 18 (4) | 1 (3) | 17 (5) | 0.63 |
| DAPT+OAC | 31 (8) | 3 (9) | 28 (8) | 0.83 |
| Statin | 222 (55) | 17 (49) | 205 (56) | 0.43 |
| Lesion characteristics | | | | |
| TASC classification | | | | |
| A/B | 163 (40) | 12 (46) | 151 (41) | 0.58 |
| C/D | 237 (59) | 19 (54) | 218 (59) | |
| CTO | 156 (39) | 18 (51) | 138 (37) | 0.10 |
| Puncture site calcification | 87 (22) | 10 (29) | 77 (21) | 0.29 |

Abbreviations: CLI, critical limb ischemia; CTO, chronic total occlusion; DAPT, dual antiplatelet therapy; OAC, oral anticoagulant therapy; PAD, peripheral arterial disease; SAPT, single antiplatelet therapy; TASC, Trans-Atlantic Inter-Society Consensus.

[a]Continuous data are presented as mean±standard deviation; categorical data are given as the count (percentage).

Meanwhile, 412 arterial accesses from the CFA, popliteal artery, distal below-the-knee arteries, brachial artery, and radial artery were performed. We excluded seven arterial accesses from the brachial artery and one arterial access from the radial artery. Hence, 366 EVT and 404 arterial access procedures were included in this study. There were 35 bleeding-associated complications (9%), all of which were hematomas. Of the 35 patients with bleeding-associated complications, all required manual re-compression and extension of the bed rest period, and 15 patients were treated with blood transfusions and treated without sequelae. No cases required operation for hemostasis. The incidence of bleeding-associated complications was significantly high among patients aged ≥80 years (BG vs. NBG, 43% vs. 25%, p<0.05). There were no significant differences in the other contributing factors between the groups. The total number of dual antiplatelet therapy (DAPT) prescriptions was 335 (83%); however, the combination of oral antiplatelet drugs did not differ between the groups (BG vs. NBG, single antiplatelet therapy (SAPT): 9% vs. 4%, p = 0.14; DAPT: 77% vs. 83%, p = 0.34; SAPT + oral anticoagulant therapy (OAC): 3% vs. 5%, p = 0.63; DAPT + OAC: 9% vs. 8%, p = 0.83). There was no significant difference in the TASC classification (BG vs. NBG, TASC A and B: 46% vs. 41%, p = 0.58; TASC C and D: 54% vs. 59%, p = 0.58), chronic total occlusion (BG vs. NBG, 51% vs. 37%, p = 0.58), and puncture site calcification (BG vs. NBG, 29% vs. 21%, p = 0.29) between the groups (Table 1).

The most common arterial access site was the CFA in 89% (antegrade cannulation; n = 194 [48%], retrograde cannulation; n = 166 [41%]); the popliteal artery was selected in 43 (11%), and a distal below-the-knee artery was used in only one (0%). The used sheath size was 83% for ≥6 Fr and 17% for a sheath with <6 Fr or only a microcatheter. The ultrasound-guided puncture was performed in 168 patients (42%), and in the remaining patients, the fluoroscopy-guided or pulsation-guided puncture was performed. The treatment procedure was stent implantation (60%) and plain-old balloon angioplasty (35%) and did not involve a drug-eluting balloon or stent. For hemostasis, protamine was used in 61%, and vascular closure devices (Angio-Seal STS-Plus™ [St. Jude Medical, Minnesota, USA]/Exoseal™ [Cordis Corporation, Miami Lakes, FL, USA]) were used in 42% of the patients. The incidence of bleeding-associated complications was significantly higher with antegrade cannulation than with retrograde cannulation of the CFA (BG vs. NBG, 48% vs. 69%, p<0.05) and in the absence of ultrasound-guided puncture (3% vs. 45%, p<0.05). The incidence of bleeding-associated complications with ultrasound-guided puncture was 1% (1/168), and the incidence without ultrasound-guided puncture was 14.4% (34/236). In this study, 236 (58%) of the total 404 cases did not undergo ultrasound-guided puncture. If ultrasound-guided puncture was performed in all cases, the incidence of complications is expected to decrease from the actual 35 cases to 2.4 cases. Contralateral cannulation of the CFA (26% vs. 43%, p = 0.09), cannulation from the popliteal artery (6% vs. 11%, p = 0.32), and distal below-the-knee arteries (BG 0% vs. 0%) were not significantly different between both groups. There was no significant difference between the groups with respect to sheath size (p = 0.35), protamine use (69% vs. 61%, p = 0.36), or vascular closure devices (40% vs. 42%, p = 0.82) (Table 2).

## Multivariable analysis of bleeding-associated complications

Table 3 demonstrates variables, including age ≥80 years, that were independently associated with increased odds of bleeding-associated complications after EVT (relative risk (RR), 2.04; odds ratio (OR), 2.82; 95% confidence interval (CI), 1.30–6.11; p<0.05). The use of real-time ultrasound-guided puncture was protective against bleeding-associated complications (RR, 0.04; OR, 0.28; 95% CI, 0.004–0.21; p<0.05).

**Table 2. Procedural characteristics[a].**

| Variables | Total | Bleeding group | Non-bleeding group | p-value |
|---|---|---|---|---|
| | **n = 404** | **n = 35** | **n = 369** | |
| Procedural characteristics | | | | |
| Access site | | | | |
| Femoral catheterization | 360 (89) | 33 (94) | 327 (89) | 0.30 |
| Antegrade cannulation | 194 (48) | 24 (69) | 170 (46) | <0.05 |
| Retrograde cannulation | 166 (41) | 9 (26) | 157 (43) | 0.09 |
| Popliteal catheterization | 43 (11) | 2 (6) | 41 (11) | 0.32 |
| Distal puncture | 1 (0) | 0 (0) | 1 (0) | |
| Sheath size | | | | 0.35 |
| ≥6 Fr | 335 (83) | 31 (89) | 304 (82) | |
| 6 Fr | 322 | 30 | 292 | |
| 7 Fr | 13 | 1 | 12 | |
| <6 Fr | 69 (17) | 4 (11) | 65 (18) | |
| Ultrasound-guided puncture | 168 (42) | 1 (3) | 167 (45) | <0.05 |
| Type of procedure | | | | |
| Stent implantation | 243 (60) | 23 (66) | 220 (60) | |
| Plain-old balloon angioplasty | 143 (35) | 7 (20) | 136 (37) | |
| Drug-eluting balloon/stent | 0 (0) | 0 (0) | 0 (0) | |
| Protamine | 248 (61) | 24 (69) | 224 (61) | 0.36 |
| Vascular closure device | 169 (42) | 14 (40) | 155 (42) | 0.82 |

[a]Categorical data are given as the count (percentage).

## Discussion

In this study, bleeding-associated complications after EVT occurred in 35 cases (9%). By re-verifying each factor, we found that bleeding events were very common among elderly people aged ≥80 years. Antegrade cannulation of the CFA and the lack of ultrasound-guided puncture were harmful as they caused bleeding events. Several previous studies have demonstrated that ischemia of the ipsilateral lower extremity is associated with hemostasis after femoral arterypuncture [18, 21]. Therefore, antegrade cannulation, which is a puncture of the lower limb on the lesion side, requires safe puncture without complications. Most of previous studies have demonstrated the usefulness of ultrasound-guided puncture; however, the rates of antegrade cannulation were as low as 0–16% [15, 16, 22, 23], which were not sufficient to evaluate the usefulness of ultrasound-guided puncture in antegrade cannulation. In this study, 48% of patients had antegrade cannulation, which indicates that ultrasound guidance is useful even in patients with a relatively high number of ipsilateral antegrade cannulations.

Previous studies have reported that patient predictors of bleeding-associated complications were increasing age and the female sex [12, 17]. Applegate RJ revealed that increasing age was

**Table 3. Multiple logistic-regression model with an odds ratio for bleeding-associated complications.**

| | Multivariate analysis | | |
|---|---|---|---|
| Variable | Odds ratio | 95% CI | p -value |
| Age ≥ 80 years | 2.82 | 1.30–6.11 | <0.05 |
| Ultrasound-guided puncture | 0.28 | 0.004–0.21 | <0.05 |

Abbreviation: CI, confidence interval.

associated with bleeding-associated complications (OR = 1.02, 95% CI 1.01–1.03, p<0.001) in 34,556 patients who underwent percutaneous cardiac catheterization via the CFA [24]. Especially in those aged ≥80 years, the risk of complication is higher [15]. In our patients, the age ≥ 80 years was a predictor of bleeding-associated complications.

Some studies reported procedural predictors of bleeding-associated complications, such as ultrasound-guided puncture [15, 16, 25], antegrade cannulation of CFA [26], sheath size [17, 27], and the use of vascular closure devices [11, 18, 28]. Although an ultrasound-guided puncture enables puncture at the calcified site to prevent bleeding-associated complications [22, 23], our study unexpectedly did not reveal an association between puncture site calcification and bleeding-associated complications. In contrast, routine ultrasound-guided punctures were protective against hematoma [15]. Hence, an ultrasound-guided puncture can not only avoid calcification but also enable puncture at an "optimal position," thereby resulting in less plaque and potential hemostasis. There are several key points related to CFA puncture: 1) avoidance of the bifurcation site of the femoral artery, especially in cases involving a high takeoff deep femoral artery, 2) avoidance of the site where there is an overlap of the branch artery and the vein, 3) avoidance of a calcified lesion or plaque burden site in the CFA, and 4) identification of the position of the femoral head. Once the puncture position had been decided, the target vessel was punctured while checking the blood vessel and puncture needle simultaneously via ultrasonography. Bleeding-associated complications were reduced by appropriate puncturing based on information obtained by ultrasonography alone.

Antegrade cannulation of the CFA is also a predictor of bleeding-associated complications. Antegrade puncture is technically more difficult than a retrograde puncture. A high puncture (above the inguinal ligament) is associated with an increased incidence of groin hematoma and retroperitoneal bleeding [26]. Yeow et al. revealed that concomitant use of echo-guide puncture when performing antegrade cannulation of the CFA might reduce bleeding-associated complications [29]. In contrast, Siracuse et al. revealed that there were no significant differences between antegrade and retrograde cannulations in the overall rates of hematoma (3% vs. 2.7%; p = 0.21) or hematoma requiring intervention (0.4% vs. 0.4%; p = 0.75) in 45,816 patients undergoing EVT [19]. For antegrade cannulation, this is controversial.

The predictors of bleeding-associated complications such as the female sex, anticoagulant use, a large sheath size, and manual compression, identified in previous studies [12–18], were not significant in this study. The implication is that the increased risk due to these factors is offset by performing an "optimal puncture" with the ultrasound-guided puncture of the CFA.

This study had some limitations. First, it was a retrospective, single-center study. Second, the sample size was small. Third, multiple surgeons with different proficiencies were involved in the puncture and hemostasis; hence, the procedures may have been performed by surgeons with different experience levels.

## Conclusions

EVT is expected to facilitate treatment for more complex and high-risk cases that could not be treated using previous techniques. Reducing complications is an important issue as high-risk patients are being increasingly targeted. Therefore, ultrasound-guided puncturing is recommended to reduce bleeding-associated complications, especially during antegrade cannulation of the CFA among elderly patients over the age of 80 years, regardless of calcified plaque at the puncture site.

## Author Contributions

**Conceptualization:** Kenji Inoue.

**Data curation:** Masayuki Shiozaki, Iwao Okai, Kenji Inoue.

**Formal analysis:** Kenji Inoue.

**Investigation:** Kentaro Fukuda, Masayuki Shiozaki, Akihisa Nishino.

**Methodology:** Shinya Okazaki, Akihisa Nishino.

**Project administration:** Kentaro Fukuda, Kenji Inoue.

**Supervision:** Shinya Okazaki, Hiroshi Tamura, Masataka Sumiyoshi, Hiroyuki Daida, Tohru Minamino.

**Writing – original draft:** Kentaro Fukuda.

**Writing – review & editing:** Shinya Okazaki.

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
