## [Decision Letter · Decision Letter 0]

27 Nov 2020

PONE-D-20-34172

Ultrasound-guided puncture reduces bleeding-associated complications, regardless of calcified plaque, after endovascular treatment of femoropopliteal lesions, especially using the antegrade procedure: A single-center study

PLOS ONE

Dear Dr Olazaki

Thank you for submitting your manuscript to PLOS ONE. After careful consideration, we feel that it has merit but does not fully meet PLOS ONE’s publication criteria as it currently stands. Therefore, we invite you to submit a revised version of the manuscript that addresses the points raised during the review process.

We look forward to receiving your revised manuscript.

Kind regards,

Xianwu Cheng, M.D., Ph.D., FAHA

Academic Editor

PLOS ONE

Additional Editor Comments:

Although the topic is interesting, as you will gather from the reviews, the referees identified substantive methodological problems/technical problems, statistical analysis, and data presentation. One reviewer has concerns the novelty of current study with raising the recent studies. The editorial broad member also concurs. You may resubmit a revised version but it will be re-reviewed and there exists no guarantee that even with revision it will necessarily be accepted.

Journal Requirements:

Reviewers' comments:

Reviewer's Responses to Questions

**Comments to the Author**

1. Is the manuscript technically sound, and do the data support the conclusions?

Reviewer #1: Partly

Reviewer #2: Yes

2. Has the statistical analysis been performed appropriately and rigorously? 

Reviewer #1: No

Reviewer #2: Yes

3. Have the authors made all data underlying the findings in their manuscript fully available?

Reviewer #1: Yes

Reviewer #2: Yes

4. Is the manuscript presented in an intelligible fashion and written in standard English?

Reviewer #1: Yes

Reviewer #2: No

5. Review Comments to the Author

Reviewer #1: 1 why is this report that ultrasound guidance is important in vascular catheterization novel given references 15 and 16 and why is this of general conceptual importance?

a. is it calcification

b. is it antegrade

just not sure why these findings are so conceptually novel as to appeal to the broad readership of

2 should it not be incidence of bleeding when US guidance is used and when not rather than was US guidance used in bleeding or not. Can the authors please provide that analysis? Otherwise we are dealing with small numbers

3 can you report relative risk not just odds ratio?

Reviewer #2: Comments::

1. The primary endpoint is not well defined - what is a "bleeding complication?" You say "main outcome measure of bleeding-associated complications was defined as bleeding at the

puncture site that required medical intervention" - what does this mean? What interventions were performed? This is poorly flushed out

2. How were these bleeding complications treated?

3. Overall tone of the text is very colloquial and should be revised

6. PLOS authors have the option to publish the peer review history of their article (what does this mean?). If published, this will include your full peer review and any attached files.

Reviewer #1: No

Reviewer #2: No

---

## [Author Response · Author response to Decision Letter 0]

26 Dec 2020

Reviewer#1

1 why is this report that ultrasound guidance is important in vascular catheterization novel given references 15 and 16 and why is this of general conceptual importance?

a. is it calcification

b. is it antegrade

just not sure why these findings are so conceptually novel as to appeal to the broad readership of

Thank you very much for your comment.

We apologize for the inadequate explanation.

In references 15 and 16, many risk factors for bleeding-associated complications have been verified, but the antegrade approach and puncture site calcification have not been verified. Although ultrasonic-guided puncture is greatly affected by artifacts in puncture site calcifications, it is novel in that it reduces the risk of bleeding-associated complications, especially in an antegrade puncture. In previous studies showing that ultrasound guide puncture is useful, many of the cases are of retrograde approaches. In modern EVT, the antegrade approach is often required, and the purpose was to retroactively confirm whether the ultrasound guide is effective in the recent EVT situation.

Therefore, we added the objective of this investigation in the Introduction section. (line 2-5, page 4)

2 should it not be incidence of bleeding when US guidance is used and when not rather than was US guidance used in bleeding or not. Can the authors please provide that analysis? Otherwise we are dealing with small numbers

Thank you very much for your comment.

The incidence of bleeding-associated complications with ultrasound-guided puncture was 0% (1/168), and the incidence without ultrasound-guided puncture was 14.4% (34/236). In this study, 236 (58%) of the total 404 cases did not undergo ultrasound-guided puncture. If ultrasound-guided puncture was performed in all cases, the incidence of complications is expected to decrease from the actual 35 cases to 2.4 cases. Therefore, we added the incidence of bleeding-associated complication in the Procedure section. (line 6-11, page 10)

3 can you report relative risk not just odds ratio?　

I would like to thank you for the constructive comment.

The following data have been added to the revised manuscript. (line 2-6, page 12)

(Age ≥ 80 years): relative risk is 2.04

(Ultrasound-guided puncture): relative risk is 0.04 

Reviewer#2

1. The primary endpoint is not well defined - what is a "bleeding complication?" You say "main outcome measure of bleeding-associated complications was defined as bleeding at the

puncture site that required medical intervention" - what does this mean? What interventions were performed? This is poorly flushed out

Thank you for your comment. 

We apologize for the inadequate explanation.

The medical interventions required are compression, an extension of the bed rest period, transfusion, and operation for hemostasis. I added it in the Endpoints section. (line 3-4, page 5)

2. How were these bleeding complications treated

Thank you for your comment.

Of the 35 patients with bleeding-associated complications, all required manual re-compression and extension of the bed rest period, and 15 patients were treated with blood transfusions and treated without sequelae. No cases required operation for hemostasis.

Therefore, we added it in the Results section. (line 3-6, page 9)

3. Overall tone of the text is very colloquial and should be revised

Thank you very much for your comment.

I have revised the manuscript in accordance with your suggestion.

---

## [Decision Letter · Decision Letter 1]

19 Jan 2021

PONE-D-20-34172R1

Ultrasound-guided puncture reduces bleeding-associated complications, regardless of calcified plaque, after endovascular treatment of femoropopliteal lesions, especially using the antegrade procedure: A single-center study

PLOS ONE

Dear Dr Okazaki

Thank you for submitting your manuscript to PLOS ONE. After careful consideration, we feel that it has merit but does not fully meet PLOS ONE’s publication criteria as it currently stands. Therefore, we invite you to submit a revised version of the manuscript that addresses the points raised during the review process.

We look forward to receiving your revised manuscript.

Kind regards,

Xianwu Cheng, M.D., Ph.D., FAHA

Academic Editor

PLOS ONE

Additional Editor Comments (if provided):

Although the original reviewer#1 has still concerned the novelty of current paper, this academic editors recommend the authors figure out the differences between previous and current studies in the discussion section.

Reviewers' comments:

Reviewer's Responses to Questions

**Comments to the Author**

1. If the authors have adequately addressed your comments raised in a previous round of review and you feel that this manuscript is now acceptable for publication, you may indicate that here to bypass the “Comments to the Author” section, enter your conflict of interest statement in the “Confidential to Editor” section, and submit your "Accept" recommendation.

Reviewer #1: (No Response)

Reviewer #2: All comments have been addressed

Reviewer #3: All comments have been addressed

2. Is the manuscript technically sound, and do the data support the conclusions?

Reviewer #1: Yes

Reviewer #2: Yes

Reviewer #3: Yes

3. Has the statistical analysis been performed appropriately and rigorously? 

Reviewer #1: Yes

Reviewer #2: Yes

Reviewer #3: Yes

4. Have the authors made all data underlying the findings in their manuscript fully available?

Reviewer #1: Yes

Reviewer #2: No

Reviewer #3: Yes

5. Is the manuscript presented in an intelligible fashion and written in standard English?

Reviewer #1: Yes

Reviewer #2: Yes

Reviewer #3: Yes

6. Review Comments to the Author

Reviewer #1: The authors have answered my queries and yet I still remain unconvinced that there is much novel here or that there is a mechanistic, conceptual or scientific aspect to this study.

Reviewer #2: All comments have been addressed. Ok to accept this paper. I have no further comments to add.

Reviewer #3: (No Response)

7. PLOS authors have the option to publish the peer review history of their article (what does this mean?). If published, this will include your full peer review and any attached files.

Reviewer #1: No

Reviewer #2: No

Reviewer #3: No

---

## [Author Response · Author response to Decision Letter 1]

9 Feb 2021

Reviewer #1: The authors have answered my queries and yet I still remain unconvinced that there is much novel here or that there is a mechanistic, conceptual or scientific aspect to this study.

Thank you very much for your comment. We have added the following information to the revised manuscript (page 12, line 12): “Several previous studies have demonstrated that ischemia of the ipsilateral lower extremity is associated with hemostasis after femoral arterypuncture [18, 21]. Therefore, antegrade cannulation, which is a puncture of the lower limb on the lesion side, requires safe puncture without complications. Most of previous studies have demonstrated the usefulness of ultrasound-guided puncture; however, the rates of antegrade cannulation were as low as 0–16% [15, 16, 22, 23], which were not sufficient to evaluate the usefulness of ultrasound-guided puncture in antegrade cannulation. In this study, 48% of patients had antegrade cannulation, which indicates that ultrasound guidance is useful even in patients with a relatively high number of ipsilateral antegrade cannulations.“

Reviewer #2: All comments have been addressed. Ok to accept this paper. I have no further comments to add.

Thank you very much for reviewing our manuscript.

Reviewer #3: (No Response)

Thank you very much for reviewing our manuscript.

---

## [Decision Letter · Decision Letter 2]

26 Feb 2021

Ultrasound-guided puncture reduces bleeding-associated complications, regardless of calcified plaque, after endovascular treatment of femoropopliteal lesions, especially using the antegrade procedure: A single-center study

PONE-D-20-34172R2

Dear Dr Okazaki

We’re pleased to inform you that your manuscript has been judged scientifically suitable for publication and will be formally accepted for publication once it meets all outstanding technical requirements.

Kind regards,

Xianwu Cheng, M.D., Ph.D., FAHA

Academic Editor

PLOS ONE

Additional Editor Comments (optional):

None.

Reviewers' comments:

Reviewer's Responses to Questions

**Comments to the Author**

1. If the authors have adequately addressed your comments raised in a previous round of review and you feel that this manuscript is now acceptable for publication, you may indicate that here to bypass the “Comments to the Author” section, enter your conflict of interest statement in the “Confidential to Editor” section, and submit your "Accept" recommendation.

Reviewer #2: All comments have been addressed

2. Is the manuscript technically sound, and do the data support the conclusions?

Reviewer #2: Yes

3. Has the statistical analysis been performed appropriately and rigorously? 

Reviewer #2: Yes

4. Have the authors made all data underlying the findings in their manuscript fully available?

Reviewer #2: Yes

5. Is the manuscript presented in an intelligible fashion and written in standard English?

Reviewer #2: Yes

6. Review Comments to the Author

Reviewer #2: Ok to accept this

paper. I have no further comments to add.

7. PLOS authors have the option to publish the peer review history of their article (what does this mean?). If published, this will include your full peer review and any attached files.

Reviewer #2: No

---

## [Editor Report · Acceptance letter]

3 Mar 2021

PONE-D-20-34172R2 

Ultrasound-guided puncture reduces bleeding-associated complications, regardless of calcified plaque, after endovascular treatment of femoropopliteal lesions, especially using the antegrade procedure: A single-center study 

Dear Dr. Okazaki:

I'm pleased to inform you that your manuscript has been deemed suitable for publication in PLOS ONE. Congratulations! Your manuscript is now with our production department. 

Kind regards, 

on behalf of

Associate Prof. Xianwu Cheng 

Academic Editor

PLOS ONE